# Diagnostic and Prognostic Value of microRNAs in Patients with Laryngeal Cancer: A Systematic Review

**DOI:** 10.3390/ncrna9010009

**Published:** 2023-01-19

**Authors:** Elisabetta Broseghini, Daria Maria Filippini, Laura Fabbri, Roberta Leonardi, Andi Abeshi, Davide Dal Molin, Matteo Fermi, Manuela Ferracin, Ignacio Javier Fernandez

**Affiliations:** 1Department of Medical and Surgical Sciences (DIMEC), Alma Mater Studiorum, Università di Bologna, 40138 Bologna, Italy; 2Division of Medical Oncology, IRCCS Azienda Ospedaliero, Universitaria Policlinico Sant’Orsola Malpighi of Bologna, 40138 Bologna, Italy; 3Department of Otorhinolaryngology—Head and Neck Surgery, IRCCS Azienda Ospedaliero, Universitaria di Bologna, Policlinico S. Orsola-Malpighi, 40138 Bologna, Italy; 4IRCCS Azienda Ospedaliero, Universitaria di Bologna, 40138 Bologna, Italy

**Keywords:** LSCC (laryngeal squamous cell carcinoma), microRNA, biomarkers

## Abstract

Laryngeal squamous cell cancer (LSCC) is one of the most common malignant tumors of the head and neck region, with a poor survival rate (5-year overall survival 50–80%) as a consequence of an advanced-stage diagnosis and high recurrence rate. Tobacco smoking and alcohol abuse are the main risk factors of LSCC development. An early diagnosis of LSCC, a prompt detection of recurrence and a more precise monitoring of the efficacy of different treatment modalities are currently needed to reduce the mortality. Therefore, the identification of effective diagnostic and prognostic biomarkers for LSCC is crucial to guide disease management and improve clinical outcomes. In the past years, a dysregulated expression of small non-coding RNAs, including microRNAs (miRNAs), has been reported in many human cancers, including LSCC, and many miRNAs have been explored for their diagnostic and prognostic potential and proposed as biomarkers. We searched electronic databases for original papers that were focused on miRNAs and LSCC, using the Preferred Reporting Items for Systematic Reviews and Meta-Analyses (PRISMA) protocol. According to the outcome, 566 articles were initially screened, of which 177 studies were selected and included in the analysis. In this systematic review, we provide an overview of the current literature on the function and the potential diagnostic and prognostic role of tissue and circulating miRNAs in LSCC.

## 1. Introduction

Head and neck squamous cell carcinomas (HNSCC) originate from the squamous cells of the oral cavity, pharynx and larynx. Laryngeal cancers (LC) encompass lesions of the mucosal lining, salivary gland tissue and cartilage. Among these, the squamous cell carcinoma (SCC) of the larynx (LSCC) is the most common form of malignant epithelial tumor, accounting for a third of HNSCC. Other rarer malignancies of LC include adenocarcinomas, sarcomas, lymphoma and neuroendocrine tumors [1,2]. 

Tobacco smoking and alcohol abuse are the main risk factors for LSCC, acting synergically in the carcinogenesis process. Other factors include gastro-esophageal reflux, diet, nutritional factors and socioeconomic status [3]. The incidence of LSCC is greater in males compared to females, usually in the sixth and seventh decades of life [4].

The most common localization for LSCC is the supraglottis in some countries (e.g., Italy, Spain, France, Finland, the Netherlands) and the glottis in others (e.g., United Kingdom, Sweden, the USA, Canada). The subglottis is the rarest localization of LSCC. 

The most common early symptoms of LSCC are hoarseness, voice changes and dysphonia (mainly with supraglottic and glottic localizations), dyspnoea and stridor with subglottic SCC. Other symptoms are dysphagia, sensation of a foreign body in the throat, hemoptysis and odynophagia. These tumors have a strong tendency to metastasize to lymph nodes, depending on the site of the primary tumor, and via blood vessels, occurring in late stages of disease. Nowadays, the staging of HNSCC, including LSCC, is based on the 8th edition of the American Joint Commission on Cancer (AJCC) [1].

Early-stage laryngeal cancers, inclusive of cT1-2N0 disease, are treated successfully with a single, locally-directed treatment modality, including local radiation therapy or surgery. Locally-advanced cancers, inclusive of cT3-4N1-3 disease, are typically treated with combination therapies. For larynx preservation, two approaches are validated: concomitant chemo/radiotherapy (CRT) and induction chemotherapy (three courses) followed by radiotherapy alone in the case of complete or partial response after induction or surgery in the case of stable or progressive disease after induction. 

Patients with massive larynx cartilage invasion (T4a), extra-laryngeal extension (T4a) or with severely impaired laryngeal function should be excluded from a larynx preservation strategy and offered upfront surgery. Outside the laryngeal-preservation strategy, the role of induction chemotherapy is not recommended, and when a non-surgical approach is preferred, the standard regimen is concomitant CRT with high-dose (100 mg/m^2^) cisplatin [5]. In the case of relapsed or metastatic disease, the first line therapy is represented by a combination strategy of chemotherapy plus pembrolizumab or cetuximab, according to the expression of PD-L1 combined positive score (CPS) >1 or <1, respectively [5]. Given this complexity, the optimal treatment approach must be discussed in a multidisciplinary team including not only the “core” specialties (radiation oncology, surgery, medical oncology) but also the disciplines involved in diagnosis and treatment support [5]. 

Despite therapeutic advances, LC has a poor survival rate (five-year overall survival of 50–80%) due to advanced stage diagnosis and high recurrence rate that occur in 25% of cases and more than 90% of recurrences occur within the first two years after primary treatment [6]. An early diagnosis of LC, early detection of recurrence and a more precise monitoring of the efficacy of different treatment modalities are currently needed to reduce the mortality. Therefore, developing effective diagnostic and prognostic biomarkers for LC is crucial to guide disease management and improve outcomes.

Among the multitude of molecular factors that can be investigated for diagnostic and/or prognostic potential, there is an increased interest in microRNAs (miRNAs). MiRNAs are small non-coding RNAs of about 22 nucleotides (nts) in length that function as guide molecules in RNA silencing. Specifically, miRNA target messenger RNAs (mRNAs) to repress mRNA translation into protein and/or to induce mRNA degradation by cleavage [7,8]. MiRNAs are fundamental in all development and biological processes [9], however an aberrant expression of miRNAs has been described and associated with many human diseases, including tumors [10,11,12,13,14]. In fact, their dysregulated expression is fundamental for carcinogenesis. MiRNAs can play different roles in tumor onset and progression based on their targets: they can act as oncogenes, named oncomiRs, when they repress tumor suppressor genes expression and sustain cancer development; or they act as tumor suppressors, by downregulating oncogenes or inhibiting tumor suppressor factors [15]. MiRNAs secreted by tumor cells into extracellular fluids, including blood and saliva, and extracellular miRNAs serve as signaling molecules to mediate cell–cell communications and can be used as potential biomarkers [16,17,18,19,20,21].

To date, many studies described dysregulated expression of miRNAs in LSCC tissue and in LSCC patient blood. Many papers have also associated miRNA expression with diagnostic and prognostic features, thus supporting their use as biomarkers. A systematic review of the literature was performed to analyze the diagnostic and prognostic role of candidate miRNAs in LSCC. In addition, we provided data about miRNAs with biological roles studied and validated in LSCC cell models.

## 2. Results

### 2.1. Study Selection

In total, our search identified 566 papers through Pubmed database searching. The first screening removed duplicated (*n* = 1), retracted (*n* = 11) and in erratum (*n* = 1) papers. Then, we excluded 129 articles for not dealing directly with the investigated issue, 34 records for full text not being available, 25 for language different to the included one and 22 for not being original papers, such as review and metanalysis. Furthermore, 343 full text articles were read and then assessed for eligibility. Of those, 166 were excluded for describing HNSCC cases without focus on the larynx (*n* = 28), for being exclusively in vitro studies (*n* = 39), for having a number of patients less than 10 or for using patient data from public databases (*n* = 50), for not quantifying miRNAs in patient samples (*n* = 43) and for other reasons, such as for not having statistically significant results or wrong miRNA sequence (*n* = 6). Finally, the review was performed on a total of 177 studies [22,23,24,25,26,27,28,29,30,31,32,33,34,35,36,37,38,39,40,41,42,43,44,45,46,47,48,49,50,51,52,53,54,55,56,57,58,59,60,61,62,63,64,65,66,67,68,69,70,71,72,73,74,75,76,77,78,79,80,81,82,83,84,85,86,87,88,89,90,91,92,93,94,95,96,97,98,99,100,101,102,103,104,105,106,107,108,109,110,111,112,113,114,115,116,117,118,119,120,121,122,123,124,125,126,127,128,129,130,131,132,133,134,135,136,137,138,139,140,141,142,143,144,145,146,147,148,149,150,151,152,153,154,155,156,157,158,159,160,161,162,163,164,165,166,167,168,169,170,171,172,173,174,175,176,177,178,179,180,181,182,183,184,185,186,187,188,189,190,191,192,193,194,195,196,197]. The flowchart of the systematic search is shown in Figure 1.

### 2.2. Characteristics of the Selected Studies

In this systematic review, the sample size of the included articles varied from 10 to 840 participants, with 10,841 cancer cases in total.

The 177 eligible articles were published between 2009 and 2022, specifically 2009 (*n* = 1), 2010 (*n* = 1), 2011 (*n* = 3), 2012 (*n* = 3), 2013 (*n* = 12), 2014 (*n* = 9), 2015 (*n* = 16), 2016 (*n* = 23), 2017 (*n* = 14), 2018 (*n* = 16), 2019 (*n* = 24), 2020 (*n* = 25), 2021 (*n* = 18) and 2022 (*n* = 12). Most of the studies were performed in China (86.4%) and about 8% of the studies were performed in Europe, mostly in Poland, Italy and Bulgaria. Regarding the study type, most of the selected articles are retrospective (97%).

The quantification of miRNA was performed mostly in cancer tissue (90.4%), and 16 studies analyzed circulating miRNA from blood compartments, including plasma, serum and extracellular vesicles (EVs).

Some papers indicated the specific tumor sampling site where the miRNA was quantified, such as superficial or deep site. Regarding tumor characteristics, 114 articles specified the T stage and 55 papers provided information about tumor localization, namely, laryngeal subsites (supraglottic, glottic and subglottic). As control group, most of the studies (80.8%) used normal laryngeal mucosa from the same patient. The main features of the 177 selected studies are shown in Table 1.

### 2.3. miRNA and LSCC

MiRNAs can be detected and quantified in tumor samples (tissue miRNAs), or in patient blood (circulating miRNAs). Analyzing the 177 selected papers, we found 383 miRNA entities reported in the study results, corresponding to 182 different tissue miRNAs and 50 different circulating miRNAs. To investigate and assess miRNA abundance of the most promising miRNAs, almost all studies used PCR-based techniques. Specifically, Wei et al. also used droplet digital PCR (ddPCR) [174], all the others used the reverse-transcription quantitative polymerase chain reaction (RT-qPCR) technique, either as the discovery or validation approach. Only Popov et al. used only a microarray approach without further PCR-based validation [32].

Some miRNAs were reported by more than one study. The most frequently studied tissue miRNA in our paper list was miR-21-5p, whose role in LSCC was described in 14 different papers [26,28,59,67,110,123,129,130,140,145,147,159,174,176]. 

Other tissue miRNAs were reported by five or more articles, including miR-145-5p [32,42,59,92,99,130,131,153,167], miR-375-3p [28,140,145,149,159,171,178,188,196], miR-155-5p [26,28,59,84,106,128,169], miR-195-5p [75,107,111,114,115,146,197], miR-106b-3p [26,127,132,145,146,196], miR-21-3p [32,42,59,97,146,168], miR-125b-5p [32,43,82,101,117,130], miR-204-5p [32,49,65,82,113], miR-205-5p [129,130,136,164,188] and miR-93-5p [32,42,97,130,148].

Only seven circulating miRNAs were reported by two or more studies, namely, miR-21-5p [57,143,174], miR-125b-5p [101,133], miR-126-3p [133,141], miR-223-3p [79,133], miR-27a-3p [133,191], miR-33a-5p [79,140] and miR-206 [43,133]. The main features of the selected miRNAs are shown in Table 2.

### 2.4. Diagnostic Potential of miRNA in LSCC

MiRNAs have been proposed as disease biomarkers that can aid cancer diagnosis. In fact, the expression of some miRNAs changes between tumor and normal tissue and is associated with diagnostic features. We collected data from miRNA differential expression between LSCC tissue and control tissue, which is usually normal mucosa. We found 69 miRNAs that are upregulated in the tumor, putative oncomiRs, and 95 miRNAs that are downregulated in tumor compared to normal mucosa, putative tumor suppressor miRNAs. Some miRNAs have a controversial biological role, because some papers reported an upregulation of the miRNA in the tumor, while other papers showed a downregulation of the same miRNA in the tumor. This is the case of miR-375-3p, where seven studies showed its downregulation in tumor tissue [28,140,145,149,159,171,188]; while two papers reported its higher expression in tumor tissue compared to controls [178,196].

In addition to being dysregulated in LSCC tumor, the dysregulation of some miRNA was studied and associated with tumor grading, T stage and presence of lymph nodal metastasis (N stage). Diagnostic association of tissue miRNAs in LSCC is shown in Appendix A. 

For circulating miRNAs, the comparison was performed between the level of miRNAs isolated from patients’ blood compartments and healthy volunteers’ blood compartments. Forty-eight circulating miRNAs seem to be associated with diagnosis, and include 34 oncomiRs, 12 tumor suppressor miRNAs and two miRNAs whose roles are controversial. Circulating miRNAs have been associated with tumor diagnosis and other diagnostic features, including tumor grading, T stage and presence of lymph nodal metastasis. The complete data are reported in Table 3. 

### 2.5. Prognostic Potential of miRNA in LSCC

MiRNA expression can be associated with cancer prognosis—usually oncomiR overexpression is associated with poor prognosis, while tumor suppressor miRNA overexpression is associated with a better prognosis. In this systematic review, we reported the miRNAs whose expression in tumor samples or blood samples has been associated with prognostic parameters, including recurrence, response to radiotherapy (RT), response to chemotherapy (CHT) and disease specific survival (DSS), which include overall survival. We found 41 tissue miRNAs and seven circulating miRNAs associated with prognosis. Among the tissue miRNAs, there are 19 oncomiRs and 21 tumor suppressor miRNAs. The prognostic role of miR-34a-5p resulted in being controversial [50,124].

For circulating miRNAs, four oncomiRs and three tumor suppressor miRNAs have been studied and reported. The main features of the miRNAs with potential prognostic relevance are reported in Table 4 and Table 5.

### 2.6. Functional Role of miRNA in Laryngeal Squamous Cell Carcinoma

Among the 177 studies included in this systematic review, about one third analyzed the functional role of the miRNAs in cell line models. We collected functional data for 47 different miRNAs from 59 original papers. miRNAs have been classified as oncomiRs (*n* = 12) or tumor suppressor miRNAs (*n* = 34) based on their biological role in cancer cells.

One miRNA, miR-26a-5p, was described with opposite roles in two studies [53,108]. For this reason, this miRNA was not included in the analysis.

We obtained a list of 57 original papers, including 11 papers describing an oncomiR; 44 describing a tumor suppressor miRNA and one paper describing both oncomiR and tumor suppressor miRNAs. A summary of these 57 studies is included in Table 6.

For each miRNA (*n* = 44), we reported the LSCC cell line where the functional role was studied, and, when available, the validated target gene(s) and the cellular processes affected by the expression of the miRNA, which include proliferation and cycle, apoptosis, metabolism, invasion and migration and response to treatment. In addition, some miRNAs were tested in vivo by cell line transplantation in mice and their association with tumor growth and/or metastasis development was reported (Table 7).

## 3. Discussion

Laryngeal squamous cell cancer (LSCC) is one of the most common malignant tumors of the head and neck region. LSCC detection at an early stage could lead to a better patient survival, however, is it frequently diagnosed at an advanced stage which, together with the high recurrence rate, leads to an overall poor survival and high mortality. Discovering new diagnostic and prognostic biomarkers for LSCC is crucial to improve disease management and outcomes.

A biomarker is an objective and measurable indicator of some biological state or condition. An ideal biomarker needs to be easily assessable, specific and sensitive to the investigated pathology, and should be translatable from research to clinic [198]. Biomarkers allow the fulfillment of the precision medicine principle, which is to guide decisions made in regard to the prevention, diagnosis and treatment of disease. In many tumor types, small non-coding RNAs, and especially miRNAs, have been extensively studied and found dysregulated, suggesting a pivotal role in cancer onset and progression and a potential utility as diagnostic or prognostic biomarkers. The first circulating miRNAs were proposed as biomarkers for cancer in 2008 for diffuse large B-cell lymphoma and prostate cancer [199,200], and then miRNA biomarkers were proposed in literature for numerous diseases, including LSCC. 

miRNA meets most of the required criteria for being an ideal biomarker. In fact, miRNAs can be easily extracted from liquid biopsy samples, they are specific for the tissue or cell type of origin and their level in biological fluids/diseased tissues vary according to disease progression [198]. Several studies showed that miRNAs can be used to differentiate cancer stages [201] and to measure therapy responsiveness [202]. 

In addition to circulating miRNAs, tissue miRNAs can also be applied to confirm the initial pathological classification and indicate the prognosis associated with cancer development [203]. The advantages of tissue miRNAs are that they can be isolated and quantified using minimally invasive methods directly from tissue samples. In addition, they are stable in formalin-fixed paraffin-embedded (FFPE) specimens [204], and tissue samples archived for long periods can still be used in retrospective studies [205]. 

The use of miRNAs as biomarkers can improve LSCC diagnosis and prognosis, as evidenced by our systematic review.

We collected association data between miRNA expression and LSCC diagnostic and prognostic features. For the diagnostic features, we found 189 tissue miRNAs whose expression is dysregulated in LSCC. Many of them show different levels depending on grading, T stage and the presence of positive lymph nodes. The majority have been described in a single study; however, 66 tissue miRNAs were reported in at least two different studies. The most studied is the oncomiR miR-21-5p, whose essential role in carcinogenesis is recognized in several cancer types [206].

For circulating miRNAs, we reported the association of 34 oncomiRs, 12 tumor suppressor miRNAs with LSCC presence or features. The most studied tumor suppressor miRNAs were miR-125b-5p and miR-126-3p, whose expressions were found to be downregulated in tumor blood compared to healthy volunteer controls, and three oncomiRs, namely, miR-21-5p, miR-27a-3p and miR-33a-3p, whose blood levels were higher in LSCC patients compared to healthy controls. The further validation of these five miRNAs could provide a novel diagnostic biomarker for LSCC. 

In addition to diagnosis, we also investigated the correlation between miRNAs and prognostic parameters, which were the presence of recurrence, response to therapy, (chemotherapy and radiotherapy) and disease-free survival. Most of the associations between tissue miRNAs and prognostic features regarded the survival: the high expression of 16 tumor suppressor miRNAs predicted a longer survival, while the high expression of 18 oncomiRs has been associated with a shorter disease specific survival (DSS). 

For circulating miRNAs, we reported the association of seven circulating miRNAs with recurrence and overall survival. Specifically, we found four oncomiRs with high expression in LSCC patients with a worst prognosis: shorter DSS for miR-1246, miR-21-5p, miR-26b-5p and miR-632, and tumor recurrence for miR-26b-5p and miR-632. Among the three tumor suppressor miRNAs, the high expression in blood compartments of miR-10a-5p and miR-126-3p have been correlated with longer DSS, while the high expression of miR-378a-3p predicted a reduced probability to develop recurrence. Each miRNA was reported only by one study, and future validations in larger cohorts will be useful to confirm their potential role in LSCC prognosis.

Some miRNAs proposed as diagnostic and/or prognostic biomarkers were also examined in in vitro models to investigate their functional role in LSCC. It is fundamental to use an appropriate cell line model able to represent the cancer properties. However, many publications, used inappropriate cell line models, such as misidentified cell lines. This is the case of the Hep-2 cell line, which was first described in 1954 as laryngeal cancer, and reported eight years later as contaminated with cervical adenocarcinoma derived HeLa cells. Nevertheless, we found many LSCC research publications that used the Hep-2 cell line [207]. In the functional section of this systematic review, we decided to exclude publications that used the Hep-2 cell line, and other cell lines that are not recognized as specific from LSCC, including TU-212, which is a generic head and neck squamous cell carcinoma cell line. The authenticated LSCC cell lines that were used in most studies for exploring the functional role of miRNAs were TU-177, AMC-HN-8 and TU686. The functional role of eight miRNAs was studied by more than one research group and thus consolidated the results, and the description was controversial only for miR-26a-5p. The seven remaining miRNAs, which are all described as tumor suppressor miRNAs, include miR-125b-5p, miR140-5p, miR-143-3p, miR-195-5p, miR-204-5p, miR-330-3p and miR-375-3p. 

MiRNA as biomarkers for various conditions, including cancer, are an impressive research field. However, they are still in their early stage of development and lack standardized procedures and reproducibility. We also reported a few discordant data from different studies. To resolve this issue, it is important to develop and use standardized protocols and test miRNAs in larger cohorts of patients to improve and strengthen the obtained results. Once validated in larger multicenter studies, miRNAs have the potential to be easily included in routine clinical practice to deliver personalized medicine to LSCC patients and improve life expectancy and quality of life.

## 4. Materials and Methods

### 4.1. Search Strategy

According to the Preferred Reporting Items for Systematic Review and Meta-Analysis (PRISMA) process [208], a comprehensive review of the MEDLINE, Embase and Web of Science databases was undertaken for literature published before October 2022, without publication year restrictions. The protocol has been registered on PROSPERO (ID: CRD42023389525). We conducted the search using different combinations of key terms and medical subject heading [MeSH Terms]. The search strategy is summarized with the following search string: “(lary*) AND (microRNA), Most Recent,,”““lary*”“[All Fields] AND (““microrna s”“[All Fields] OR ““micrornas”“[MeSH Terms] OR ““micrornas”“[All Fields] OR ““microrna”“[All Fields])”.

Reference lists from screened articles and review articles were also searched. The cross-references from selected studies were further searched for additional articles. Articles identified through this search were screened and evaluated using identical study selection criteria. Titles and abstracts were independently screened for relevance by the seven authors of this study (FIJ, BE, FDM, LR, FL, AA, DMD), while disagreements were resolved through discussions with the other authors (MFm, MFe).

### 4.2. Inclusion and Exclusion Criteria

Studies fulfilling the following criteria were included: original research studies in which the expression of miRNAs in LSCC tissue or blood was investigated, and studies examining the role of miRNA in therapeutic drug or radiation resistance. Studies were included if reporting validation of the data with a control group. For studies extracting miRNAs from cancer tissue, we included them if the control group consisted either of non-carcinoma tissue from the same patient (extracted from other head and neck mucosa) or from different patients (extracted from the larynx). In case of circulating miRNAs extracted from blood, we included the study if the control group consisted of healthy patients. For circulating miRNAs, all blood compartments (serum, plasma and circulating vesicles) were considered for inclusion in the study.

Investigations on exogenous regulatory RNAs or non-original research articles, such as review articles, conference proceedings, editorials and book chapters, were excluded. Studies not written in English, studies with less than 10 patients, lacking a control group, studies not reporting clear descriptions of the cancer histology (i.e., studies including other conditions or not being specific about squamous cell cancer of the larynx), studies not reporting the miRNA extraction method and studies extracting miRNAs from cellular lines and not from LSCC tissue or LSCC patient’s blood, were excluded. Studies which did not focus on the clinical aspects of the extracted miRNA(s) (i.e., on its diagnostic value, treatment outcome predictive value or prognostic value) were excluded as well.

All the selected studies were included in the statistical analysis of the clinical parameters. The studies were further included in the functional analysis of single miRNA pathological mechanisms if they reported validation on cellular lines of laryngeal squamous cell cancer. Studies with validation on Hep2 cells were excluded from the functional analysis.

### 4.3. Data Extraction

Seven independent authors (FIJ, BE, FDM, LR, FL, AA, DMD) retrieved information from the eligible articles following the inclusion and exclusion criteria, and data were collected on a standardized data sheet that included: first author name and publication year, biological specimen, sample size, control group, cell line validation, miRNAs expression, measurement method and the outcomes of interest: diagnostic and prognostic. The results of the five independent reviewers were compared, and disagreements on search strategy, article inclusion and data extraction were resolved by an independent reader.

### 4.4. Study Quality Assessment

The methodologic quality of the included studies was evaluated independently by six authors, using the Quality Assessment of Diagnostic Accuracy Studies-2 (QUADAS-2) criteria [209]. If disagreements between the six reviewers occurred, they would discuss them to achieve a consensus or consult with the seventh reviewer (IJF). 

### 4.5. Statistical Analysis

The statistical analysis was conducted with SPSS 26.0 software (SPSS, IBM), while final tables were obtained with JASP version 0.16.30. Descriptive analysis of the variables was carried out. Variables included in the analysis were the laryngeal site of extraction of miRNAs, blood compartment for circulating miRNAs, features of the laryngeal cancer (tumor stage, grading and nodal involvement), treatment response (radiotherapy and chemotherapy) and prognostic parameters (recurrence and disease specific survival). The association of miRNAs’ expression with clinical parameters were displayed in contingency tables. The type of association and the type of expression (downregulation or upregulation) of the miRNAs were summarized. Mean values were reported for continuous variables with normal distribution. Median values were reported for variables without normal distribution. Confidence intervals were set at 95%.

### 4.6. miRNA Nomenclature

Over the years, there were changes in mature miRNA nomenclature and not all papers used the updated names reported in microRNA official database, such as miRBase. The actual nomenclature introduced the -3p and -5p suffixes convention for mature miRNA naming as a substitute for the star (*) symbol for the less predominant form. When the determination of the identity of mature miRNAs was possible, we converted the miRNA name found in original papers and used the MIRBASE v.22 updated name.

### 4.7. Study Selection of Functional Analysis

Among the identified papers for the systematic review, we selected those that performed functional analysis of miRNAs in a correct cell line model, such as a cell line model recognized as being derived from LSCC. The exclusion parameters for functional analysis were: (i) not performing functional analysis, (ii) not using an appropriate cell line model or (iii) not studying the specific effect of a miRNA, but the combination of it with other molecular factors, such as long non-coding RNAs. 

## 5. Conclusions

Literature on miRNAs in LSCC is quite extensive, specifically that on tissue miRNAs, while limited data are available for circulating miRNAs. Many of the miRNAs suitable for a clinical application as diagnostic or prognostic biomarkers were reported singularly or by a few studies, rarely with controversial results. Furthermore, the findings generally lack validation, which increases the uncertainty in view of potential clinical applications. Therefore, large clinical validation studies on the identified miRNAs are required to select clinically relevant miRNAs and translate those results into the clinical practice. 

## Figures and Tables

**Figure 1 ncrna-09-00009-f001:**
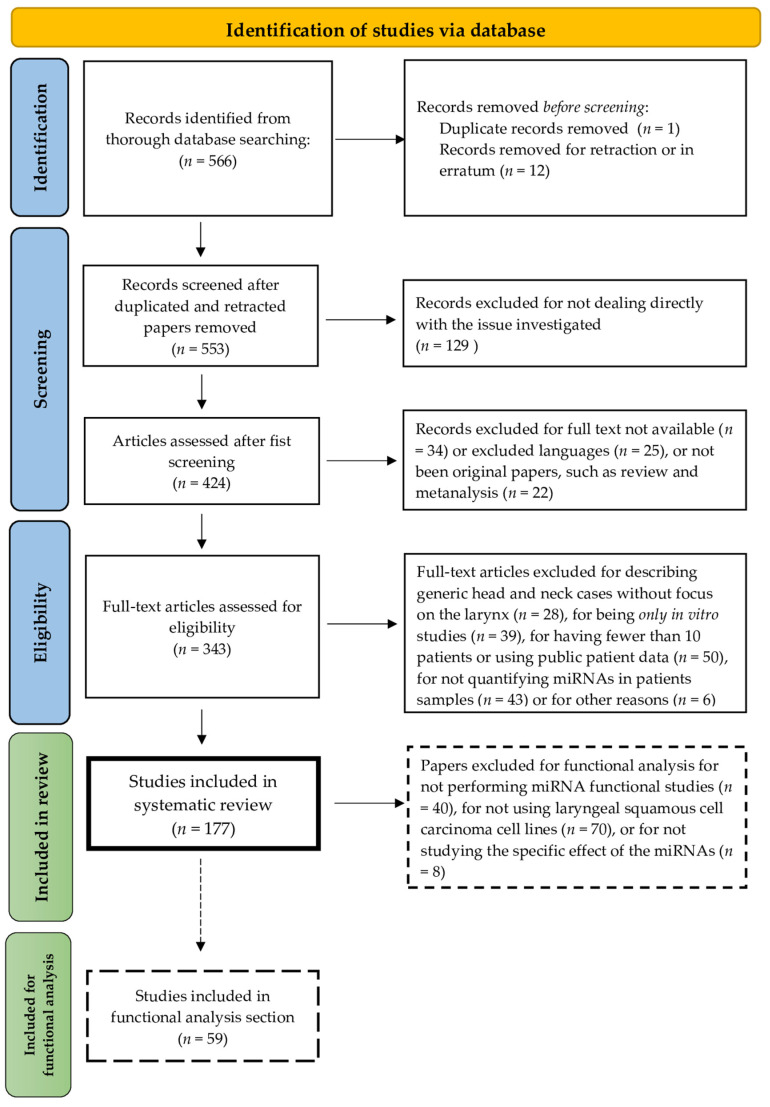
Flow chart showing the steps of the systematic review of the literature. Of 566 papers, 177 original papers were selected in this systematic review. For functional analysis, a further screening was performed and 59 original papers were obtained.

**Table 1 ncrna-09-00009-t001:** Description of general features of the 177 selected studies in this systematic review.

Case Description		
	Frequency *	Percentage
General features		
Publication Date		
2009–2011	5	2.8
2012–2014	24	13.6
2015–2017	53	30
2018–2020	65	36.7
2021–2022	30	16.9
Total	177	100
Country		
Brazil	2	1.1
Bulgaria	4	2.3
China	153	86.4
Greece	1	0.6
Egypt	1	0.6
Italy	4	2.3
Japan	1	0.6
Poland	5	2.8
Turkey	6	3.3
Total	177	100
Study type		
Prospective	5	2.8
Retrospective	172	97.2
Total	177	100
Site of miRNA extraction		
Cancer tissue	160	90.4
Peripheral blood	11	6.2
Both	6	3.4
Total	177	100
Blood Compartment		
Serum	7	43.8
Plasma	6	37.5
Extracellular vesicles (EVs)	2	12.5
Missing (Peripheral blood)	1	6.2
Total	16	100
Tumor sampling site		
Superficial and deep	8	4.5
Deep	2	1.1
Not defined	167	94.4
Total	177	100
Tumor features		
T Stage		
T1–T2(early)	3	1.7
T3–T4 (advanced)	12	6.8
T1–T4 (any stage)	99	55.9
Not defined	63	35.6
Total	177	100
Localization (laryngeal subsites)		
All sites	29	16.4
Supraglottic	5	2.8
Supraglottic and glottic	20	11.3
Glottic	1	0.6
Not defined	122	68.9
Total	177	100
Control Group		
Normal laryngeal mucosa (same patient)	143	80.8
Normal laryngeal mucosa (same patient) and healthy blood (another person)	4	2.3
Normal laryngeal mucosa (another patient)	13	7.3
Other (including healthy blood from another patient)	15	8.5
Not defined	2	1.1
Total	177	100

* Frequency corresponds to number of papers.

**Table 2 ncrna-09-00009-t002:** Description of miRNA data of the 177 selected studies in this systematic review.

miRNA Description		
	Frequency *	Percentage
Type of microRNA		
Tissue miRNA	182	78.4
Circulating miRNA	50	21.6
Total	232	100
Tissue miRNA		
Described by more than 10 papers	2	1.1
Described by 5–9 papers	9	4.9
Described by 2–4 papers	54	29.7
Described by only 1 paper	117	64.3
Total	182	100
Circulating miRNA		
Described by 2–3 papers	7	14
Described by only 1 paper	43	86
Total	50	100

* Frequency corresponds to number of papers.

**Table 3 ncrna-09-00009-t003:** Circulating miRNAs associated with diagnostic features in LSCC.

Circulating miRNA	Dysplasia	LSCC	Grading	T stage	N Stage	Number of Paper(s) *	Cohort Numerosity (Range) #	Ref(s)
Downregulated								
miR-125b-5p	0	2 ↓	0	0	0	2	124 (60–64)	[101,133]
miR-126-3p	0	2 ↓	1↓	0	0	2	102 (38–64)	[133,141]
miR-133a-3p	0	1 ↓	0	0	0	1	66	[79]
miR-145-5p	0	1 ↓	0	0	0	1	66	[79]
miR-150-5p	0	1 ↓	0	0	0	1	64	[133]
miR-19a-3p	0	1 ↓	0	0	0	1	64	[133]
miR-192-5p	0	1↓	0	0	0	1	64	[133]
miR-203-3p	0	1 ↓	0	0	0	1	64	[133]
miR-218-5p	0	1 ↓	0	0	0	1	64	[133]
miR-25-3p	0	1 ↓	0	0	0	1	64	[133]
miR-451a	0	1 ↓	0	0	0	1	64	[133]
miR-601	0	1 ↓	0	0	0	1	64	[133]
Upregulated								
let-7a-5p	0	1 ↑	0	0	0	1	66	[79]
miR-106b-5p	0	1 ↑	0	0	0	1	64	[133]
miR-10a-5p	0	1 ↑	1 ↑	1 ↑	0	1	236	[57]
miR-122-5p	0	1 ↑	0	0	0	1	66	[79]
miR-1246	0	1 ↑	0	0	0	1	61	[61]
miR-130a-3p	0	1 ↑	0	0	0	1	64	[133]
miR-141-3p	0	1 ↑	0	0	0	1	66	[79]
miR-149a-5p	0	1 ↑	1 ↑	1 ↑	1 ↑	1	66	[79]
miR-155-5p	0	1 ↑	0	1 ↑	1 ↑	1	840	[169]
miR-182-5p	0	1 ↑	0	0	0	1	66	[79]
miR-191-5p	0	1 ↑	0	0	0	1	64	[133]
miR-195-5p	0	1 ↑	0	0	0	1	64	[133]
miR-196a-5p	0	1 ↑	0	0	0	1	22	[39]
miR-19b-3p	0	1 ↑	0	0	0	1	64	[133]
miR-20a-5p	0	1 ↑	0	0	0	1	64	[133]
miR-21-5p	1 ↑	3 ↑	0	2 ↑	1 ↑	3	404 (52–236)	[57,143,174]
miR-221-3p	0	1 ↑	0	0	0	1	60	[154]
miR-26b-5p	0	1 ↑	0	1 ↑	0	1	59	[116]
miR-27a-3p	0	2 ↑	0	0	0	2	128 (64–64)	[133,191]
miR-27b-3p	0	1 ↑	0	0	0	1	64	[133]
miR-30b-5p	0	1 ↑	0	0	0	1	64	[133]
miR-31-3p	0	1 ↑	0	0	0	1	22	[39]
miR-31-5p	0	1 ↑	0	0	0	1	66	[79]
miR-320a-3p	0	1 ↑	0	0	0	1	64	[133]
miR-328-3p	0	1 ↑	0	0	0	1	64	[133]
miR-33a-5p	0	2 ↑	0	0	0	2	76 (10–66)	[79,140]
miR-375-3p	0	1 ↑	0	0	0	1	64	[133]
miR-378a-5p	0	1 ↑	1 ↑	1 ↑	0	1	384	[189]
miR-424-5p	0	1 ↑	0	0	0	1	22	[39]
miR-484	0	1 ↑	0	0	0	1	64	[133]
miR-485-3p	0	1 ↑	0	0	0	1	66	[79]
miR-632	0	1 ↑	1↑	1↑	1↑	1	162	[63]
miR-93-5p	0	1 ↑	0	0	0	1	64	[133]
miR-941	0	1 ↑	0	0	0	1	59	[33]
Other								
miR-206	0	1 ↑,1 ↓	0	0	0	2	64↑ 68↓	[133,193]
miR-223-3p	0	1 ↑,1 ↓	0	0	0	2	64↑ 66↓	[79,133]

* Numbers correspond to the number of studies reporting the association with the diagnostic parameter. ↑: upregulated miRNA. ↓: downregulated miRNA. # Cohort numerosity indicates the number of patients, and when two or more papers reported the same miRNA, the minimum and the maximum number of patients is indicated inside brackets.

**Table 4 ncrna-09-00009-t004:** Tissue miRNAs associated with prognostic features in LSCC patients.

Tissue miRNA	Recurrence	Response to RT	Response to CHT	DSS	Number of Paper(s) *	Cohort Numerosity (Range) #	Ref(s)
Tumor suppressor miRNA							
miR-101-3p	0	0	0	1	1	80	[156]
miR-107	0	0	1	0	1	30	[22]
miR-1205	1	0	0	1	1	44	[70]
miR-140-5p	0	0	0	1	1	56	[83]
miR-143-3p	0	0	0	2	2	112 (52–60)	[90,93]
miR-145-5p	0	0	0	2	2	320 (132–188)	[92,99]
miR-147a	0	0	0	1	1	45	[24]
miR-149-5p	0	0	0	1	1	143	[179]
miR-154-5p	0	0	0	1	1	104	[100]
miR-195-5p	1	0	0	3	3	402 (98–182)	[114,115,197]
miR-204-5p	0	0	0	1	1	20	[49]
miR-22-3p	1	0	0	1	1	49	[76]
miR-29c-3p	0	0	0	1	1	66	[74]
miR-300	0	0	0	1	1	133	[190]
miR-375-3p	0	0	0	1	1	46	[159]
miR-497-5p	0	0	1	0	1	38	[30]
miR-518a-3p	0	0	1	0	1	60	[47]
miR-519a-3p	0	0	0	1	1	96	[135]
miR-655-3p	0	0	1	0	1	105	[23]
miR-766-5p	0	0	0	1	1	60	[40]
miR-873-5p	0	0	1	0	1	28	[35]
OncomiRs							
miR-1246	0	0	0	1	1	61	[61]
miR-144-3p	1	0	0	0	1	60	[42]
miR-146a-5p	0	0	0	1	1	33	[50]
miR-17-5p	0	0	0	1	1	39	[66]
miR-196b-3p	0	0	0	1	1	79	[91]
miR-196b-5p	0	0	0	1	1	113	[103]
miR-19a-3p	0	0	0	1	1	83	[142]
miR-20b-5p	0	0	0	1	1	105	[31]
miR-210-3p	0	0	0	1	1	60	[42]
miR-21-5p	0	0	0	2	2	92 (46–46)	[147,159]
miR-23a-3p	0	0	0	1	1	52	[155]
miR-26a-5p	0	0	0	1	1	56	[53]
miR-27a-3p	0	0	0	1	1	62	[191]
miR-296-5p	1	1	0	0	1	34	[157]
miR-301a-3p	0	0	0	1	1	120	[163]
miR-34b-5p	0	0	0	1	1	33	[50]
miR-34c-5p	2	0	0	2	2	133 (43–90)	[110,151]
miR-93-5p	0	0	0	1	1	60	[42]
miR-9-5p	0	0	0	1	1	103	[150]

* Numbers correspond to the number of studies reporting the association with the diagnostic parameter. # Cohort numerosity indicates the number of patients, and when two or more papers reported the same miRNA, the minimum and the maximum number of patients is indicated inside brackets.

**Table 5 ncrna-09-00009-t005:** Circulating miRNAs associated with prognostic features in LSCC patients.

Circulating miRNA	Recurrence	DSS	Number of Paper(s) *	Cohort Numerosity #	Ref(s)
OncomiR					
miR-1246	0	1	1	61	[61]
miR-21-5p	0	1	1	236	[57]
miR-26b-5p	1	1	1	59	[116]
miR-632	1	1	1	162	[63]
Tumor suppressor miRNA					
miR-10a-5p	0	1	1	236	[57]
miR-126-3p	0	1	1	38	[141]
miR-378a-3p	1	0	1	384	[189]

* Numbers correspond to the number of studies reporting the association with the diagnostic parameter. # Cohort numerosity indicates the number of patients, and when two or more papers reported the same miRNA, the minimum and the maximum number of patients is indicated inside brackets.

**Table 6 ncrna-09-00009-t006:** General information about the 59 papers with miRNA functional analysis.

miRNA Functional Analysis Papers Description			
	OncomiRs	Tumor Suppressor miRNAs	Total
Papers	12 *	44 *	56 *
miRNAs	12 (26.1%)	34 (73.9%)	46 (100%)
In vitro models			
1 LSCC cell line	7 (12.5%)	24 (42.9%)	31 (55.4%)
2 LSCC cell lines	4 (7.1%)	15 (26.8%)	19 (33.9.%)
3 LSCC cell lines	1 (1.8%)	4 (7.1%)	5 (8.9%)
4 LSCC cell lines	-	1 (1.8%)	1 (1.8%)
Total	12 (21.4%)	44 (78.6%)	56 (100%)
LSCC cell lines			
AMC-HN-8	3 (3.4%)	20 (22.7%)	23 (26.1%)
TU-177	4 (4.6%)	19 (21.6%)	23 (26.2%)
TU686	3 (3.4%)	11 (12.5%)	14 (15.9%)
SNU889	3 (3.4%)	5 (5.7%)	8 (9.1%)
Others	5 (5.7%)	15 (17%)	20 (22.7%)
Total	18 (20.5%)	70 (79.5%)	88 (100%)
Validated Target			
Yes	8 (17.4%)	29 (63%)	37 (80.4%)
No	4 (8.7%)	5 (10.9%)	9 (19.6%)
Total	12 (26.1%)	34 (73.9%)	46 (100%)
Cellular processes			
Cell proliferation and cycle	7 (6.3%)	38 (34.6%)	41 (40.9%)
Apoptosis	1 (0.9%)	18 (16.4%)	19 (17.3%)
Invasion and migration	9 (8.2%)	30 (27.3%)	39 (35.5%)
Metabolism	-	1 (0.9%)	1 (0.9%)
Drug sensitive	-	3 (2.7%)	3 (2.7%)
NA		3 (2.7%)	3 (2.7%)
Total	17 (15.4%)	93 (84.6%)	110 (100%)

* 11 papers describing only an oncomiR; 43 describing only a tumor suppressor miRNA and one paper describing both oncomiR and tumor suppressor miRNA.

**Table 7 ncrna-09-00009-t007:** miRNA’s functional role in LSCC cell line model.

miRNA	LSCC cell line(s)	Target(s)	Cellular Processes	Ref(s)
OncomiRs				
miR-148a-3p	TU686	DNMT1	Proliferation, invasion and migration	[182]
miR-155-5p	AMC-HN-8, TU-177	-	Proliferation, invasion and migration	[84]
miR-196a-5p	JHU-011	-	Proliferation, tumor growth and metastasis	[131]
miR-205-5p	SNU899	-	Invasion and migration	[188]
miR-27a-3p	AMC-HN-8, TU686	SMAD4	Proliferation, invasion and migration, tumor growth	[191]
miR-301a-3p	TU-177	SMAD4	Proliferation, apoptosis, invasion and migration	[163]
miR-302b-3p	SNU899, SNU1066, SNU1076	TGFBR2	Invasion and migration	[52]
miR-340-3p	TU-177	ELK1	-	[38]
miR-503-5p	AMC-HN-8	PDCD4	Proliferation, invasion and migration	[112]
miR-744-3p	SNU899, SNU1076,	PDCD4, PTEN	Invasion and migration	[177]
miR-941	TU686, FD-LSC-1	-	Proliferation, invasion and migration	[33]
miR-98-5p	TU-177	HMGA2	-	[77]
Tumor suppressor miRNAs				
let-7c-5p	TU-177, FD-LSC-1	-	Proliferation, apoptosis, invasion and migration	[34]
miR-107	TU686	CACNA2D1	Proliferation, invasion and migration	[72]
miR-125b-5p	AMC-HN-8, TU-177	STAT3, HK2	Proliferation, invasion and migration, metabolism	[101,117]
miR-136a-5p	TU686	RAP2C	-	[64]
miR-139-3p	TU-177, HNO210,	KDM5B	Proliferation, apoptosis, migration and invasion	[36]
miR-140-5p	AMC-HN-8, TU-177, SNU46, UM-SCC-10B	FGF9, NFAT5	Proliferation, apoptosis, migration and invasion	[68,192]
miR-141-3p	AMC-HN-8	HOXC6	Proliferation, migration and invasion, tumor growth and metastasis	[73]
miR-143-3p	TU-177, TU686, SNU889	k-Ras, MAGE-A9	Proliferation, apoptosis, invasion and migration, tumor growth	[90,93]
miR-145-5p	TU-177	FSCN1	Proliferation and cycle, apoptosis, invasion and migration	[92]
miR-195-5p	AMC-HN-8, TU-177	cyclin D1, ROCK1, DCUN1D1	Proliferation and cycle, apoptosis, migration and invasion	[75,107,111,114]
miR-204-5p	TU-177, AMC-HN-3, HN-10	ZEB1, FOXC1, SEMA4B	Proliferation, apoptosis, migration and invasion, tumor growth	[49,65,113]
miR-206	SNU899, TR-LCC-1	SOX9	Proliferation, apoptosis, migration and invasion	[193]
miR-218-5p	AMC-HN-8	CCAT1	Proliferation, migration and invasion	[118]
miR-22-3p	AMC-HN-8	NLRP3	Migration and invasion	[76]
miR-24-3p	AMC-HN-8	XIAP	Proliferation, apoptosis, response to treatment (radiosensitivity)	[152]
miR-330-3p	AMC–HN–8, TU686, SNU-899, FD-LSC-1	Tra2β, SLC7A11	Proliferation, apoptosis, invasion and migration	[56,58]
miR-340–5p	AMC-HN-8, TU177, TU686	NEAT1	Proliferation, invasion and migration, tumor growth	[54]
miR-363-3p	TU-177	Mcl-1	Proliferation, apoptosis, invasion and migration	[98]
miR-365a-3p	UM-SCC-17A	-	Proliferation	[44]
miR-375-3p	SNU899, SNU46	IGF1R	Proliferation, apoptosis, invasion and migration	[149,188]
miR-381-3p	AMC-HN-3	NASP	Proliferation, invasion and migration	[71]
miR-384	TU686	WISP1	Proliferation, apoptosis	[89]
miR-449a	HNO210	-	Proliferation	[28]
miR-4497	TU686, UM-SCC-17A	GBX2	Proliferation, apoptosis, migration and invasion	[102]
miR-4735-3p	AMC-HN-8, TU-177, UM-SCC-17A	TNFAIP3	Proliferation, invasion and migration	[51]
miR-486-3p	TU686, AMC-HN-8	LASP1	Proliferation and cycle, apoptosis, migration and invasion	[37]
miR-493-3p	AMC-HN-8	-	Proliferation, apoptosis	[69]
miR-506-3p	TU-177	YAP1	Proliferation and cycle, apoptosis, migration and invasion	[87]
miR-518a-3p	TU-177, TU686	SPATS2L	Response to treatment (cisplatin sensitivity)	[47]
miR-613	TU686	PDK1	Proliferation, invasion and migration	[109]
miR-625-5p	AMC-HN-8, TU-177	SOX4	Proliferation, invasion and migration	[88]
miR-654-3p	TU-177, FD-LSC-1	USP7	-	[55]
miR-766-5p	AMC-HN-8, TU686	HMGA2	Invasion and migration	[40]
miR-873-5p	AMC-HN-8, TU-177	-	Proliferation, apoptosis, migration and invasion, response to treatment (cisplatin sensitivity)	[35]

## Data Availability

This study did not generate a data archive. All data are already included in the manuscript.

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
