# Peer review of "Diagnostic and Prognostic Value of microRNAs in Patients with Laryngeal Cancer: A Systematic Review"

_ncrna, 2023, doi:10.3390/ncrna9010009_

Round 1

Reviewer 1 Report

This manuscript aimed to summarize the diagnostic and prognostic value of both cellular as well as circulating microRNA in laryngeal cancer. Authors have presented original research studies on expression of miRNAs in tissues or blood together with some original articles with functional studies on cell line models. Authors have also mentioned contradictory studies and the reasons to exclude them from this study. Manuscript was also easy to read and understand.

I have few suggestions and comments which could be added.

1. miRNA as biomarker is still in the growing phase because most of studies are done on smaller number of cohorts. Some study has contradictory results. Authors may include numbers of patients included in each study and used techniques to quantify gene expression, so the reliability and consistency of gene expression of the study could be helpful to the readers.

2  2. I have not found mentioning salivary circulating miRNA in the manuscript, it may be because there are not yet any study on LSCC, but I think, it will be nice to cite in introduction part that saliva could be also used for miRNAs study together with blood plasma or serum.

Overall, in this systematic review the authors have summarized most of the recent studies so far on the role of microRNA as diagnostic and prognostic biomarker. Hence, in my opinion, it could be published.

Reviewer 2 Report

LSCC is a leading malignancy in the head and neck region and early detection indeed is very important but still in need of improvement. In this manuscript, the authors summarized recent achievements in miRNAs as biomarkers in LSCC via a systemic review manner. I appreciate their work and the quality without any doubt is sound in the field. However, although I seldom comment on the resolution of the figure as I myself am not even good at it, still, I have to point out that the authors are required to make certain betterment in this regard. Otherwise, it is a very well-constructed paper. I really appreciate it. 
